# Evaluation of in situ tissue-engineered arteriovenous grafts suitable for cannulation in a large animal model
Paul J. Besseling[1,2], Wojciech Szymczyk [3], Martin Teraa [1,2], Raechel J. Toorop[2], Paul. A. A. Bartels[3], Boris Arts[3], Rob C. H. Driessen [3], Arturo M. Lichauco[3], Hidde C. Bakker[1], Joost O. Fledderus[1], Gert J. de Borst[2], Patricia Y. W. Dankers [3], Carlijn V. C. Bouten [3] & Marianne C. Verhaar [1] ✉

The sustainability of vascular access for hemodialysis is limited by frequent interventions and the inability of synthetic grafts to self-heal. Tissue engineering offers a solution through biodegradable grafts that remodel into autologous tissue. Here we assess electrospun polycarbonate-bis urea (PC-BU) vascular scaffolds (6mm-inner-Ø), reinforced with 3D-printed polycaprolactone coils, in a goat model, and compared them to expanded polytetrafluoroethylene (ePTFE) controls. The tissue-engineered grafts were repeatedly cannuled starting two weeks after implantation and were evaluated using computed tomography and histological analyses. By 12 weeks, the PC-BU grafts remodel into autologous tissue while maintaining structural integrity, maintaining integrity without dilations, ruptures, or aneurysms. Cannulation does not interfere with scaffold degradation or neo-tissue formation. Although the patency rate is lower for the PC-BU grafts (50%) compared to ePTFE (100%), the engineered grafts exhibit a self-healing response not seen in ePTFE. These findings demonstrate the potential of PC-BU tissue-engineered grafts as healing, functional vascular access solutions for hemodialysis, supporting cannulation during tissue transformation.

Repeated access to the circulation is essential for providing successful hemodialysis. Currently, the preferred vascular access to facilitate this is an autologous arteriovenous (AV) fistula. However, their use is limited by pre-existing conditions, prior surgeries, unsuitable autologous vessels, or maturation failure[1]. Although currently available synthetic alternatives like expanded polytetrafluoroethylene (ePTFE) can be used immediately without the need for maturation, they lack the healing properties of autologous tissue and are susceptible to infection, intimal hyperplasia, and thrombosis[2,3].

Recurrent complications in synthetic vascular access grafts are related to the repeated insertion of large-bore needles required for hemodialysis[4,5]. The disruption of the graft wall caused by repetitive punctures results in loss of graft integrity[6]; hematomas, (pseudo) aneurysm formation, and eventual graft failure[7]. The non-healing nature of ePTFE grafts further exacerbates these problems by providing niches for bacterial accumulation and impeding the host's immune response to infections[8,9]. Furthermore, the mismatch in compliance between the rigid synthetic grafts and the outflow veins

frequently leads to hemodynamic stress and the development of venous neointimal hyperplasia[10,11].

Vascular tissue engineering (TE) may offer many benefits over traditional synthetic grafts, as TE vascular access grafts, most importantly, are able to heal. Furthermore, they can adapt to local conditions and flow patterns, resembling natural vessels, and allow the immune system to manage infections effectively. Our proposed solution aims to bridge the gap between autologous AV fistulas and synthetic grafts by leveraging in situ tissue engineering to create a functional, living vascular access. We recently reported that a synthetic, biodegradable supramolecular electrospun scaffold fully transforms into a living, compliant vascular graft within 12 weeks after implantation in a large animal AV-shunt model[12]. The grafts could be successfully cannuled after termination. These grafts consisted of Polycarbonate-Bis Urea (PC-BU), a polymer with supramolecular properties that was also previously successfully applied in vitro[13,14] and in vivo[12,15,16]. Due to its supramolecular nature, this material enables a modular design, allowing for the incorporation of BU-functionalized additives to fine-tune mechanical properties[14] and facilitate functionalization with bioactive

[1]Department of Nephrology and Hypertension, Regenerative Medicine Centre, University Medical Centre Utrecht, Utrecht, the Netherlands. [2]Department of Vascular Surgery, University Medical Centre Utrecht, Utrecht, the Netherlands. [3]Department of Biomedical Engineering, and Institute for Complex Molecular Systems, Eindhoven University of Technology, Eindhoven, the Netherlands. ✉e-mail: M.C.Verhaar@UMCUtrecht.nl

components[17], anti-microbial[18], and/or anti-fouling additives[19]. These highly adaptable characteristics could help address key challenges associated with fully synthetic grafts, such as infection, intimal hyperplasia, remodeling, and thrombosis. However, we observed no improved patency of these grafts compared to the gold standard ePTFE grafts[12]. This may have been related to the presence of a 3D-printed anti-kinking coil made of polycaprolactone (PCL), which tends to fragment under continuous (hemodynamic) stress[20]. The resulting coil fragments protruding inside the lumen may increase the risk of thrombosis.

In this proof-of-concept study, we pursued a modular approach, combining the previously developed PC-BU electrospun grafts with a newly designed 3D-printed coil with supramolecular properties. This was achieved by blending PCL with bis urea-modified PCL to mitigate fracturing induced by the high crystallinity of PCL and enhance patency.

We then examined the in vivo performance of this adapted TE vascular access graft in goats while concurrently assessing the impacts of cannulation on the various stages of remodeling, starting at 2 weeks after implantation. For this purpose, we evaluated scaffold resorption, mechanical behavior, vascular neo-tissue formation, healing capacity, and graft patency over a 12-week follow-up period.

## Results

### Synthetic scaffold design

The luminal layer was made of electrospun PC-BU (PC-BU: $M_n = 21$ kg/mol) with a thickness of $450 \pm 50$ μm. Directly onto this luminal electrospun layer, an anti-kinking spiral made of PCL/PCL-BU 60:40 w/w (PCL: $M_n = 45$ kg/mol, PCL-BU: $M_n = 2.7$ kg/mol) (Fig. 1A) was 3D printed, with a thickness of $0.52 \pm 0.06$ mm and an interspacing of $2.73 \pm 0.26$ mm (Fig. 1B). On top of that, the outer layer, with a thickness of $104 \pm 20$ μm, made of electrospun PC-BU was applied, sandwiching the 3D-printed spiral between the luminal and outer electrospun layer. The spiral design displayed local kinking resistance and dimensional stability in the lumen; no folds were observed. No morphological changes were observed after 3D printing on top of the electrospun layer. The fiber diameter at the luminal side of the bare PC-BU grafts was $3.6 \pm 0.4$ μm. The fiber diameter of the outer layer made of PC-BU grafts was $3.6 \pm 0.5$ μm. The diameter of the lumen of the grafts was $5.9 \pm 0.2$ mm.

Stress-strain curves for 3D-printed coils (Fig. 1C) of different polymers and their mixtures revealed that the modulus of pure PCL ($107.4 \pm 0.8$ MPa) was two orders of magnitude higher than that of the 80:20 w/w PCL-BU:PCL mixture ($8.1 \pm 0.06$ MPa), pure PCL-BU ($7.9 \pm 0.09$ MPa), and PC-BU ($5.5 \pm 0.03$ MPa), all of which did not provide sufficient resistance to kinking. Increasing the PCL content to 40% led to an increase in the modulus to $42.1 \pm 0.4$ MPa. The thermal properties of the materials revealed

that in the second heating run for the mixtures, only a melting transition was observed between 50 and 55 °C, which is similar to pure PCL. The enthalpy of this transition increases with higher PCL content, with 20% having 5.2 J/g, 40% 23.7 J/, and pure PCL 67.2 J/g. Pure PCL-BU has a melting transition at 117.5 °C, corresponding to the melting of the bis urea hard blocks. In the first heating run, the 60:40 w/w PCL-BU:PCL showed a second melting transition at 116.1 °C corresponding to the bis urea stacks. Upon melting, these BU do not form crystals again, which might be caused by the PCL hindering their interaction by interacting with the PCL chains of the PCL-BU copolymer. The glass transition was around −60 °C for all the polymers and mixtures containing PCL. These results indicate that the PCL chains crystallize and dominate the interactions. The amount of crystallization increases with higher PCL content.

### Survival and graft patency

Grafts were explanted in 6 animals at 12 weeks (PC-BU $n = 8$; ePTFE $n = 2$) and 2 animals at 4 weeks (PC-BU n = 3 [cannulated n = 2, non-cannulated n = 1]; ePTFE n = 1) due to early termination because of bilateral stenosis in the jugular outflow veins, causing brain edema. No deformations (dilation, aneurysms, and/or ruptures) were observed at explanation, and PC-BU explants showed remodeling from synthetic to neo-tissue (Fig. 2A). No complications were found in sham-operated vessels. Coil fragmentation was not observed at any point during the study. Angiograms at 4 weeks showed that 62.5% of TE grafts and 66.7% of ePTFE grafts had a patent lumen (Fig. 9B). However, at this time point, two animals were prematurely sacrificed due to complications, resulting in the explantation of one ePTFE graft, two cannulated TE grafts, and one non-cannulated TE graft. At 8 weeks, 66.7% of ePTFE grafts and 56% of TE grafts remained patent. By 12 weeks, 45% of TE grafts and 66.7% of ePTFE grafts were still patent (Fig. 2B). Occlusions occurred in all groups, indicating no specific impact of cannulation on graft patency. Specifically, 2 out of 8 involved occlusion of the jugular vein itself, while 6 out of 8 cases showed occlusion of the graft at the anastomosis site. No stenosis was observed near the arterial anastomoses. Wall thickness was observed to increase over time (Fig. 2C), from the initial 0.6 mm at implantation to $0.76 \pm 0.06$ mm on average after 4 weeks increased to $2.64 \pm 0.18$ mm after 12 weeks. The wall thickness was also thicker when compared to the native CCA ($0.78 \pm 0.04$ mm, $p < 0.01$). The wall thickness of the implanted ePTFE grafts remained stable at an average of $1.46 \pm 0.06$ mm during the study The lumen diameter of the TE grafts, initially $5.9 \pm 0.2$ mm, decreased over time, measuring $4.8 \pm 0.3$ mm at 4 weeks and $3.8 \pm 0.3$ mm at 12 weeks, whereas the average lumen diameter of the implanted ePTFE grafts maintained an average lumen diameter of $5.6 \pm 0.3$ mm throughout the study (Fig. 2D). Scanning electron microscopy images taken of the luminal surface (Fig. 2E)

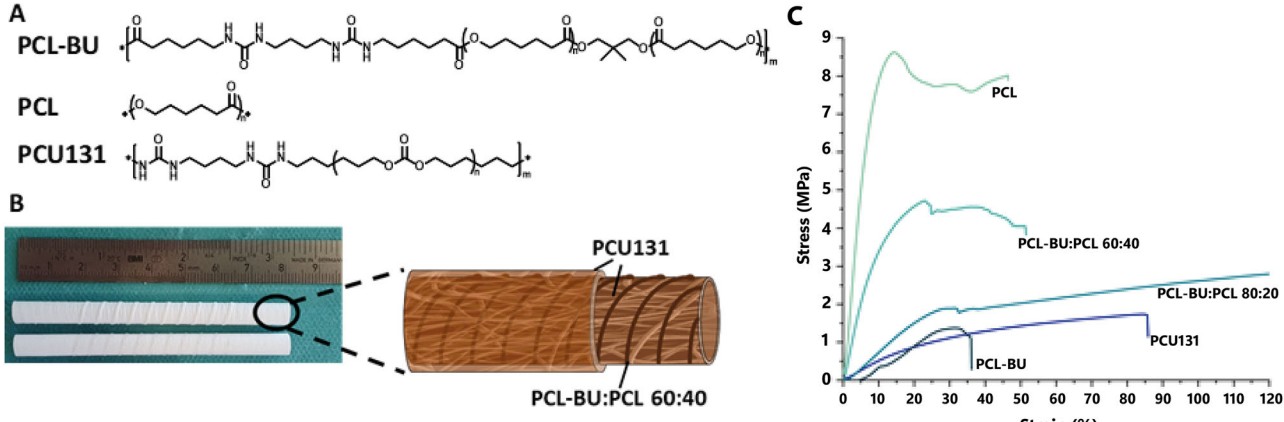

**Fig. 1 | Graft composition.** Chemical structures of PCL-BU, PCL, and PC-BU (**A**). Image of the grafts, including coils as well as a schematic representation (**B**). Stress-strain curves of individual polymers and polymer mixtures (**C**).

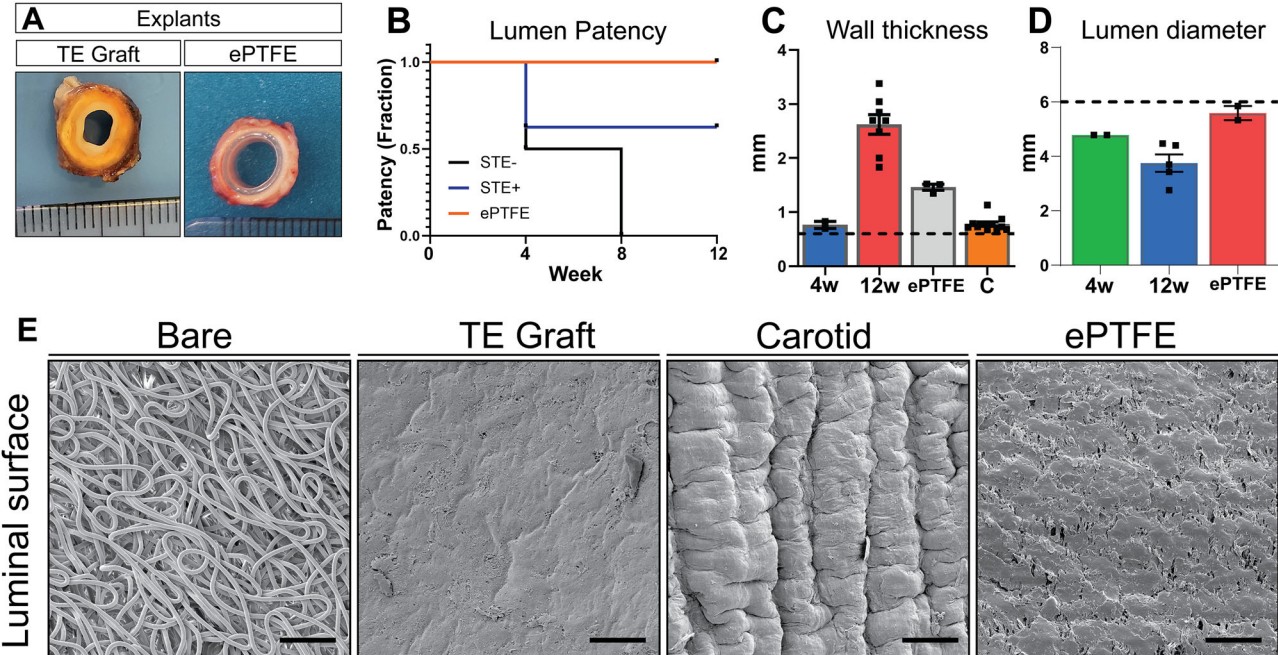

**Fig. 2 | Graft explant characteristics.** Cross-sections of the grafts and ePTFe control at explantation show tissue formation and remodeling (scale 1 mm) (**A**). Patency of TE graft vs ePTFE (**B**). The wall thickness of the TE graft at 4 weeks (n = 2) and 12 weeks (n = 8) compared to carotid (n = 11) and ePTFE controls (n = 3), dashed line at 0.6 mm is the wall thickness of the scaffold before implantation (**C**). The lumen diameter (mm) of the TE graft at 4 weeks (n = 2) and 12 weeks (n = 5) compared to ePTFE controls (n = 2), dashed line, a 6 mm is the lumen diameter of the scaffold before implantation (**D**). Scanning electron microscopy images of the luminal surface of the grafts (bare and 12 weeks), native carotid, and ePTFE control. Scale bars represent 100 µm (**E**). Data are shown as mean ± SEM.

showed the fiber structure of the initial implant compared to the smooth surface of the TE graft explanted at 12 weeks.

### Material degradation, tissue formation, and moduli

At 4 weeks most of the scaffold was intact, with cells infiltrating and ECM deposited (Fig. 3A, C), and fibers showed damage on the outer surface of the graft (Fig. 3D). After 12 weeks the scaffold was no longer intact and isolated regions with fiber material were only found in the adventitial layer of the TE vessel (Fig. 3B, E), with the remaining fibers in an advanced state of degradation (Fig. 3F). Most of the vessel consisted of newly formed tissue (Fig. 3B, G). The mechanical behavior of explanted grafts was assessed by co-axial ring testing (Fig. 3H; Supplementary Fig. 2C). During multiple stretches, grafts returned to their original morphology (Supplementary Movie 1). The stress-strain curves of TE grafts changed from 4 to 12 weeks to increasingly resemble that of the native carotid (Fig. 3I). However, the native carotid samples were still more elastic, with a more sigmoid increase in strain/displacement. The ePTFE grafts were much stiffer and at 50% strain showed a significant ($p > 0.01$) higher stress of 74.64 ± 0.63 MPa, compared to the TE grafts with an average of 16.5 ± 2.54 MPa and 6.9 ± 3.25 MPa for the 4- and 12-week explants, respectively. The native carotid was measured at 50% strain to have a stress of 0.48 ± 0.15 MPa, significantly different ($p < 0.01$) from the 4-week explants and much stiffer ePTFE.

### Cannulation

Grafts were cannulated with a 15G dialysis needle in sedated goats (Fig. 4, Supplementary Movie 2). Cannulation was guided by ultrasound to ensure cannulation of the graft and not the proximal CCA or IJV, and was performed at two, six, and ten weeks after implantation. Timely hemostasis was achieved after a maximum of 5 min with slight gauze pressure and no deformations, (pseudo)aneurysms, and/or ruptures occurred.

### AV-shunt angiogram

Two weeks after the first cannulation (4 weeks after implantation), when most of the scaffold was still intact, 3D angiograms showed observable puncture sites in the TE grafts (Yellow arrow - Fig. 5A). No puncture sites were observed in angiograms at later time points (8 and 12 weeks) when a large part of the vessels consisted of neo-tissue. Persistent damage was found in the ePTFE grafts at all time points (Yellow arrows - Fig. 5B), and flow indentations were visible around puncture sites. Irrespective of the graft type, enlarged outflow jugular veins with stenosis around the venous anastomoses were observed, which did not resolve over time (Blue arrows - Fig. 5).

### Effect of cannulation on graft characteristics and ECM formation

H&E-staining (Fig. 6A) showed an even distribution of cells throughout the graft from 4 weeks onward. Masson's Trichrome staining (Fig. 6B) (*used to stain connective tissue and muscle fibers)* indicated no difference in connective tissue deposited between non-cannulated and cannulated grafts over time. Wall thickness was significantly ($p < 0.01$) increased at 12 weeks for both cannulated and non-cannulated groups (2.94 ± 0.45 mm and 2.52 ± 0.20 mm respectively versus 0.61 ± 0.50 mm before implantation), and no influence of cannulation was observed (Fig. 6C). Both groups showed similar stress (Fig. 6D), and no significant (p: 0.49) difference was observed between cannulated (8.51 ± 4.45 MPa) and non-cannulated (2.9 ± 1.09 MPa) TE grafts at 50% strain (Fig. 6E). Gene expression analysis of collagen types 1 (Col 1) and 3 (Col 3) revealed no significant differences between cannulated and non-cannulated grafts (Supplementary Fig. 3), with Col 1 (p: 0.88) and Col 3 (p: 0.73) showing similar expression levels. However, when comparing grafts to the carotid artery, significant differences were observed for Col 1, with both cannulated ($p < 0.01$) and non-cannulated ($p < 0.01$) grafts showing lower expression levels compared to the carotid artery. For Col 3, no significant differences were found between the grafts and the carotid artery (cannulated p: 0.79; non-cannulated p: 0.35).

### Effect of cannulation on vascular tissue formation

Smooth muscle marker αSMA was present from 4 weeks (9.60 ± 2.06% - Supplementary Fig. 4) onward, showed no significant difference at 12 weeks between cannulated (17.4 ± 4.47%) and non-cannulated (10.4 ± 5.10%)

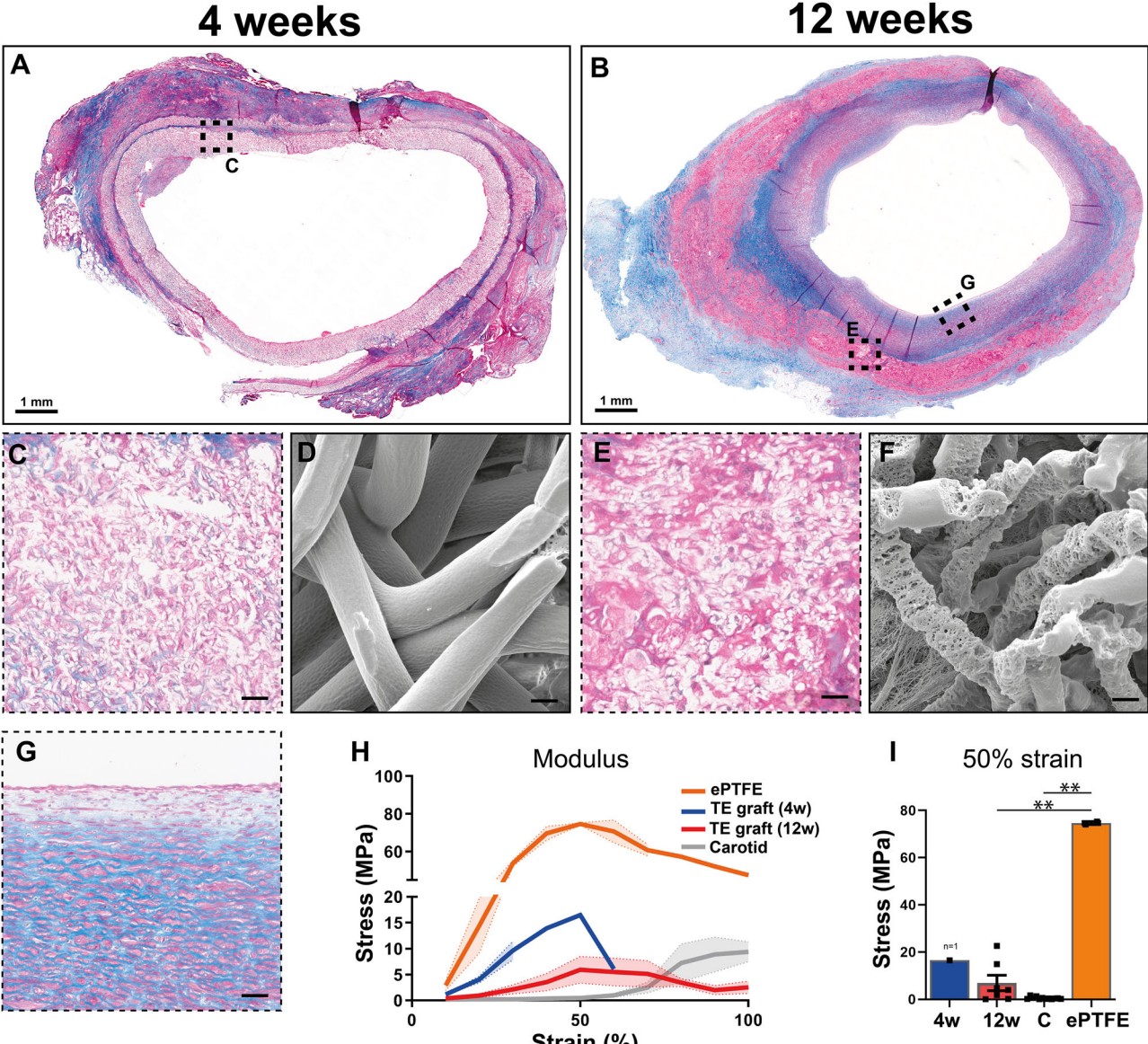

**Fig. 3 | Tissue formation and Scaffold resorption.** Transverse sections stained with Masson's trichrome (cytoplasm (pink), Muscle fibers (red), collagen (blue), and nuclei (dark blue), explanted at 4 (**A**) and 12 weeks (**B**). Zoomed in of graft at 4 weeks (**C**) and at 12 weeks neo-intima (**G**) and remnant scaffold material (**E**). Fiber integrity at 4 weeks (**D**) and 12 weeks explantation of the remaining scaffold (**F**). Average stress (MPA) - strain plots derived from ring tests (**H**) and stress at 50% strain for the various explants at 4 (n = 1) and 12 weeks (n = 7), native carotid (**C**) (n = 12) and ePTFE controls (n = 2) (**I**). Scale bars (**A, B**) (1 mm), **C, E, G** (25 μm) (**D, F**) (2.5 μm). Data is shown as mean ± SEM. *$p < 0.05$, **$p < 0.01$. Kruskal–Wallis with Dunn's post-hoc (**I**).

grafts, and was similar to the native carotid artery (26.26 ± 2.90%) (Fig. 7A, F). The presence of cells positive for the intermediate contractile marker calponin (Fig. 7B, G, - Supplementary Fig. 4B) increased over time (4 weeks 4.88 ± 0.49%) and was significantly ($p < 0.05$) lower when compared to the native CCA (15.18 ± 1.61%) but showed no effect of cannulation at 12 weeks (non-canulated: 7.72 ± 6.68%, cannulated: 7.21 ± 4.72%). Mature contractile markers, i.e., Myosin Heavy Chain (MYH), showed an increase over time from 0.58 ± 0.08% at 4 weeks to 2.10 ± 2.01% (non-cannulated) and 3.04 ± 1.04 (cannulated) at 12 weeks, all significantly ($p < 0.01$) lower compared to the native CCA (14.35 ± 2.87%) (Fig. 7C, H, - Supplementary Fig. 4C). Likewise, the surface area positive for the mature vascular marker elastin was significantly lower ($p < 0.01$), independent of time point or cannulation, compared with the native CCA, with 0.9 ± 0.03% at 4 weeks and 0.62 ± 0.14% and 0.62 ± 0.29% at 12 weeks (Fig. 7D, I, - Supplementary Fig. 4D). All TE graft sections at 12 weeks showed the presence of an endothelial cell layer demonstrated by eNOS staining (Fig. 7E, -

Supplementary Fig. 4C). Gene expression analysis of alpha-smooth muscle actin (αSMA) revealed no significant differences between cannulated (0.1674 ± 0.0824) and non-cannulated (0.0688 ± 0.0125) grafts (p: 0.9282). However, both graft types showed significantly lower αSMA expression compared to the carotid artery (8.092 ± 2.861), with cannulated vs. carotid ($p < 0.0001$) and non-cannulated vs. carotid (p: 0.0008) (Supplementary Fig. 3).

In occluded grafts, Neointimal hyperplasia was predominantly observed at the venous anastomotic region, characterized visually by a reduced lumen diameter, abundant connective tissue deposition (Fig. 8B), a high presence of αSMA-positive cells (Fig. 8B), and a chronic inflammatory response primarily driven by macrophages (Fig. 8C).

## Discussion
Our study shows successful repeated cannulation of an in situ remodeling biodegradable supramolecular electrospun vascular scaffold during the

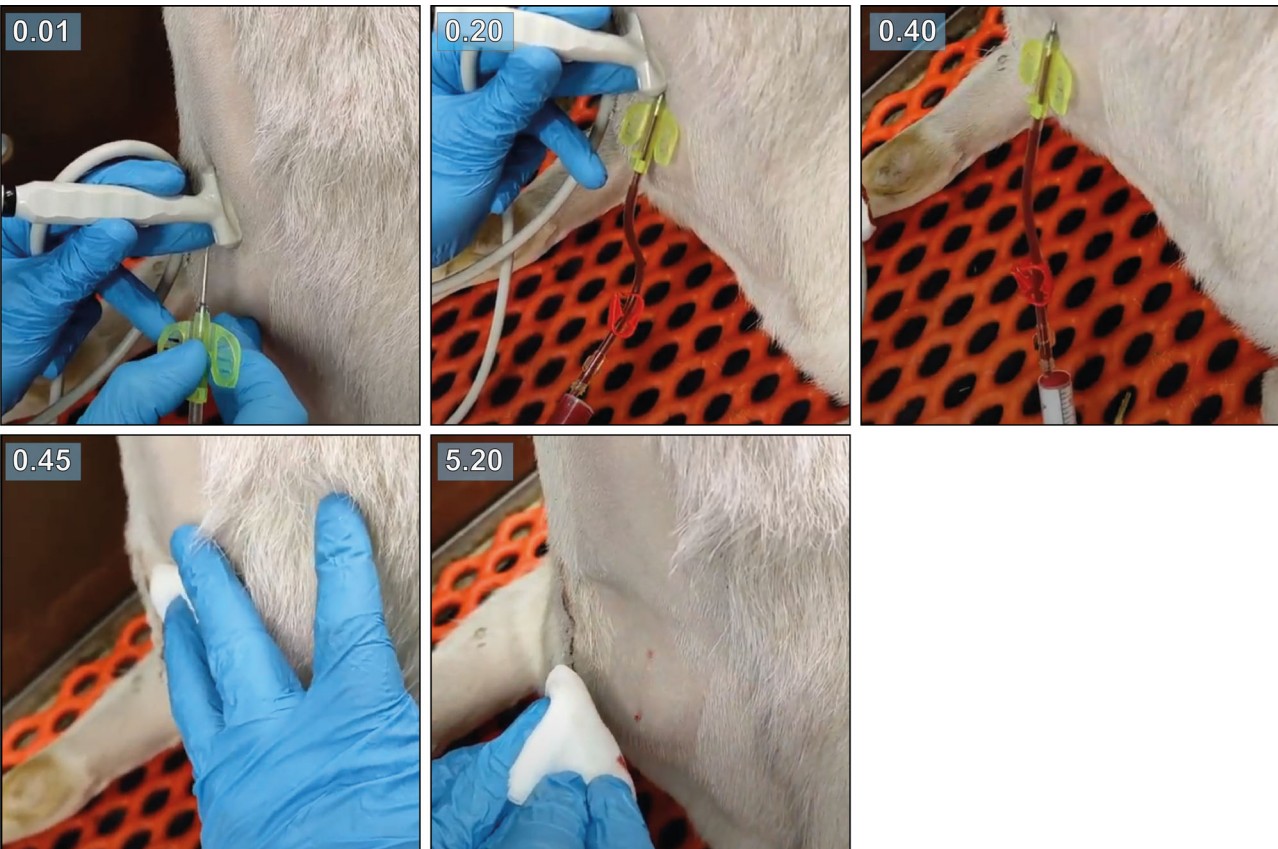

**Fig. 4 | Cannulation of the graft.** Timestamped images of ultrasound-guided cannulation with a 15G dialysis needle in an awake goat at week 2 after implantation. Hemostasis was achieved after 5 min with slightly pressured gauze.

transition from synthetic graft to autologous tissue. Our newly designed 3D-printed anti-kinking coil, made of a mix of PCL and supramolecular PCL-BU, showed compliance with the native vasculature, and we observed no fragmentation as previously observed with a coil consisting of pure PCL[12]. This improvement enables loop implantation in future applications, as it combines anti-kinking functionality with a fully degradable coil that resists fragmentation under strain or during degradation. However, these improved characteristics did not translate into increased patency of the TE grafts at 12 weeks, compared to our previous study[12]. Cannulation was performed as early as 2 weeks after implantation, and TE grafts showed self-healing after cannulation. No effects of cannulation on PC-BU scaffold breakdown or neo-tissue formation were observed.

Contrary to the TE grafts, cannulation of ePTFE grafts caused persistent damage at the cannulation sites, which has been reported to relate to loss of graft integrity[6]; hematomas, (pseudo)aneurysm formation, and eventual graft failure[7]. Previously, we successfully cannulated grafts after termination, i.e., after full remodeling into autologous tissue[12]. Here, we studied the effects of cannulation and subsequent healing throughout the transformation process, starting at 2 weeks after graft implantation, covering the three distinct phases: fully synthetic, during transformation, and fully autologous. In the TE grafts, we only observed injury at the site of cannulation on the angiogram at 4 weeks after implantation (2 weeks after the first cannulation), the time point where most of the polymer scaffold is still present and limited neo-tissue formation has occurred. At later time points, these cannulation sites were not visible on CT, suggesting adequate remodeling and self-healing capacity of the graft. In addition, despite repeated access with a large-bore needle, no deformations, (pseudo) aneurysms, or ruptures occurred. Even in the absence of anticoagulation therapy, after the initial implantation, no graft occlusion occurred during or directly after cannulation. No loss of mechanical stability was observed with

cannulation. Hemostasis was achieved within 5 min with slight external gauze compression; well within the margins of venous or prosthetic AV grafts used clinically[21].

Although our previous study[12], using a similar PC-BU electrospun graft, showed the full transformation of a synthetic, biodegradable supramolecular electrospun scaffold into a living and compliant vascular graft within 12 weeks, patency rates were lower as compared to ePTFE controls. We proposed that this could be associated with fracturing of the reinforcing PCL spiral, causing fragments to protrude into the lumen. The newly developed 3D-printed coil, composed of a supramolecular PCL blend, exhibited no fragmentation and degraded synchronously with the electrospun scaffold layers, thereby mitigating a known complication associated with stiff coil materials. However, despite these improvements in coil design, patency rates of the in situ TE grafts did not improve at 12 weeks compared to our previous study. Our findings suggest other causes for the loss of patency, such as progressive stenosis at the venous anastomosis, driven by (neo)intimal hyperplasia (IH), as indicated by the high presence of connective tissue, αSMA-positive cells, and macrophages. This is consistent with the primary mode of AV graft failure clinically[22], and previous reports on AV graft failure in the goat model[12,23,24]. The location and development of stenosis were found to be similar to our previous study, primarily occurring at the venous anastomosis[12].

Considering that mechanical stimulation and inflammation, central to neo-tissue formation, are also key mediators of IH, the prevalence of stenosis may be linked to the observed rapid tissue formation[25–27]. Stenosis was not associated with cannulation, suggesting that the transformation process itself is the primary cause. Furthermore, the rapid degradation of the scaffold materials shifts the mechanical burden to the newly formed tissue, making mechanical stimulation a major driver of tissue remodeling and adaptation. This is reflected by increased vascular

**Fig. 5 | Healing of cannulation sites in the graft.** Computed tomography scan images of grafts (**A**) and ePTFE controls (**B**) at 4, 8, and 12 weeks after cannulation at 2, 6, and 10 weeks. Visible cannulation sites are indicated by yellow arrows. Dilated jugular vein indicated by blue arrows. Scale bars (1 cm).

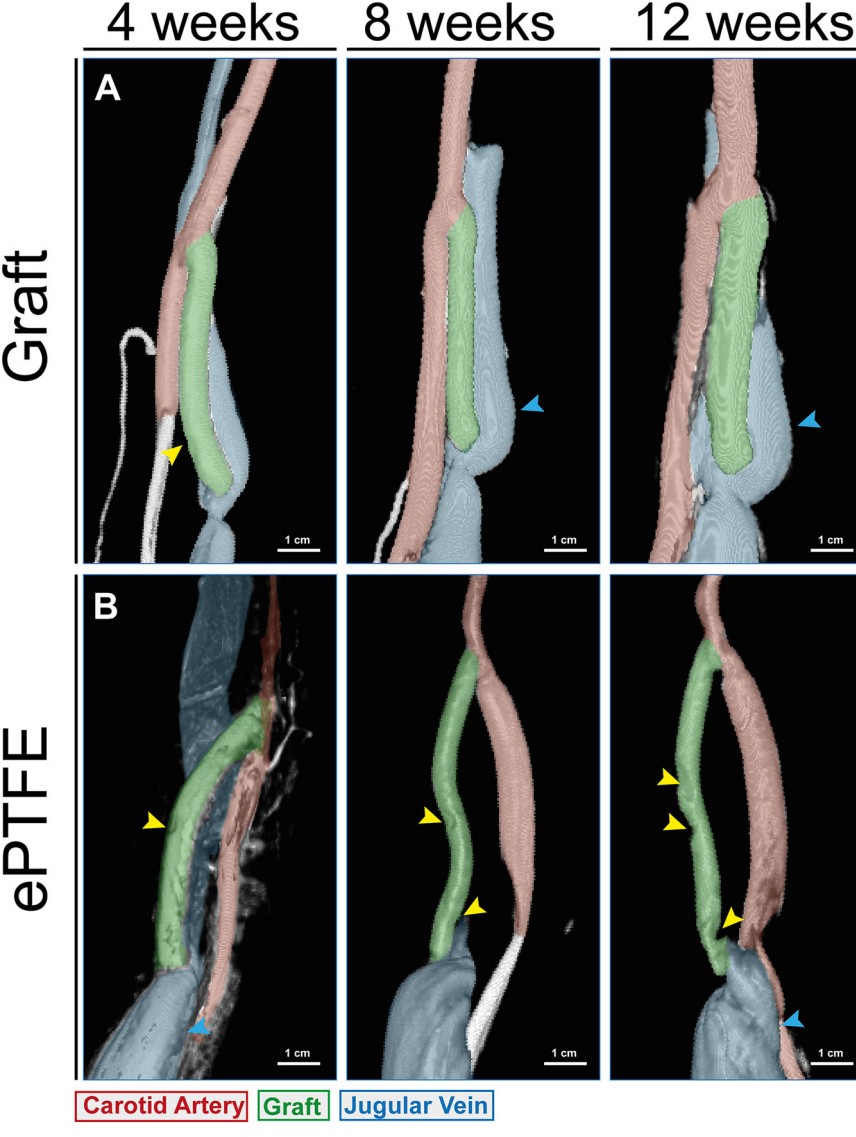

wall thickness and reduced lumen diameter. Despite the rapid formation of ECM, the newly formed vessels lack sufficient mature ECM components, such as elastin fibers, which are essential for providing mechanical strength and preventing dilation under physiological pressures. This deficiency is compensated by an increase in vascular tissue volume, primarily composed of structurally weaker collagen types, as reflected by elevated collagen III expression[28]. This compensatory mechanism resembles the remodeling observed in veins during AV fistula maturation[29]. Consistent with previous research, we expect that as the neovessels mature and gain mechanical integrity, wall thickness may eventually decrease and stabilize[26]. Additionally, we observed signs of mechanical stress at the venous anastomosis sites, indicated by enlargement of the outflow vein, which may contribute to long-term graft narrowing[30,31]. This is also implicated by the fact that bilateral stenosis of the jugular vein was observed in two animals, affecting both the ePTFE and TE grafts. This may suggest a similar adverse effect on the native vasculature, as is also observed in native AV fistula[23,32]. Additionally, as fully synthetic grafts lack the mechanical drivers of tissue formation due to their non-degradable nature and inability to support ingrowth, IH initially is limited to the anastomotic regions in these grafts. Importantly, we did not observe thrombosis in the grafts occluded in this study, an encouraging result, given that thrombosis is a major complication in clinical vascular access, particularly with synthetic grafts.

We used a clinically relevant large animal that accurately represents human arterial dimensions, complex hemodynamics, and systemic processes such as neo-tissue formation and scaffold breakdown[33]. Furthermore, goats are known to exhibit a similar rate of intimal hyperplasia (IH) and stenosis[23],[24] as observed in humans. However, our model also has limitations. We used healthy goats without renal failure, and thus, we cannot exclude the potential impact of uremic conditions on the in situ TE process. Although large animal uremic models exist, translating these findings to systemic human conditions remains challenging[34]. Additionally, the primary aim of this study was to evaluate whether cannulation and the subsequent response of the graft were feasible at different stages of graft remodeling, rather than to replicate the effect of the clinical frequency of cannulation, where patients typically undergo cannulation multiple times per week. Here, marking puncture sites could provide valuable insights into the healing process by allowing for a direct assessment of tissue response at specific locations. However, we faced challenges in tracking the cannulation sites post-explantation. The rapid healing response led to integration of the puncture sites into the surrounding tissue, making them macroscopically undetectable. Additionally, complete resorption of the implanted material eliminated any residual structural cues that could have assisted in locating the original puncture sites. Ultrasound-guided cannulation in awake goats was primarily aimed at achieving accurate graft access, positioned between the jugular vein and carotid artery. This approach, however, introduced variability in the location of puncture sites

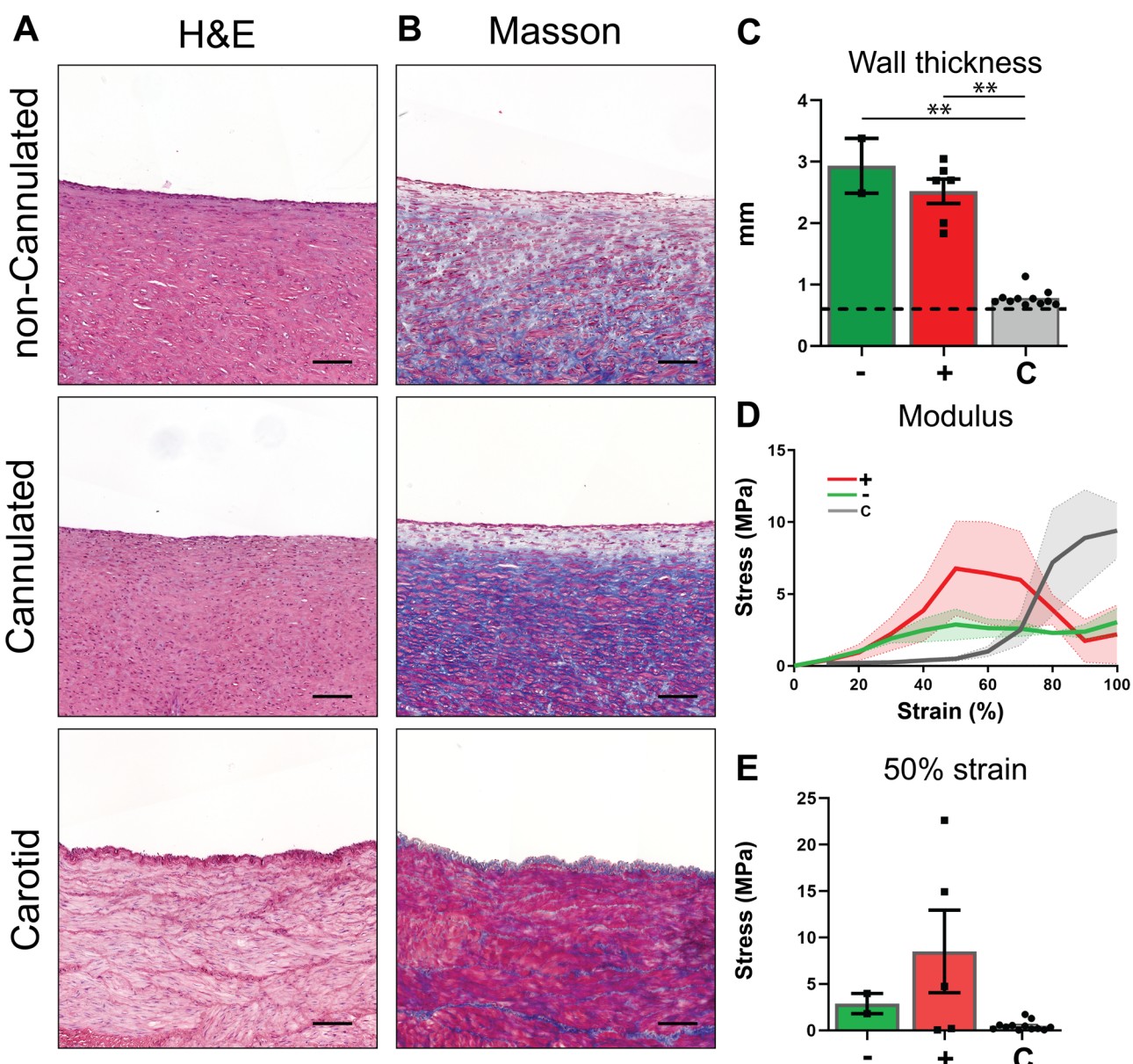

**Fig. 6 | Impact of cannulation on tissue formation.** Histological evaluation on representative transverse sections of the center part of the grafts, explanted at 12 weeks, with the native carotid artery as control, stained with H&E (**A**), Masson's Trichrome (**B**, **C**) (cytoplasm (pink), Muscle fibers (red), collagen (blue), and nuclei (dark blue). Wall thickness in mm (**C**), with the wall thickness of the graft implant (0.6 mm) represented by a dashed line. Stress (MPA) / Strain plot of ring test displacement (**D**) and stress at 50% strain (**E**). Non-cannulated (−)(n = 2), Cannulated (+)(n = 5), Carotid Artery (**C**)(n = 12). Scale bars 100 µm. Data are shown as mean ± SEM. *p < 0.05, **p < 0.01. Kruskal–Wallis with Dunn's post-hoc (**C**, **E**).

and hindered consistent identification after explantation. To enable more precise localization and monitoring of site-specific tissue responses in future studies, permanent or long-term tracking methods should be employed to allow for the selection of graft sections for histological analysis. Furthermore, an important clinical question remains regarding how well these grafts respond to interventions such as balloon angioplasty, which is frequently used to maintain patency in current vascular access options.

To further evaluate and translate our technology for clinical applications, it will be essential to conduct longer-term follow-up studies that assess not only the resorption of the implanted material, as demonstrated, but also the subsequent tissue maturation phase. Particularly given that ePTFE grafts demonstrate primary patency rates of only 40–50% at one year[21,35]. Moreover, future studies should focus on understanding why intimal hyperplasia (IH) predominantly occurs near the venous anastomosis, rather than the arterial side, in in situ *TE* vascular access grafts. Investigating the interplay between hemodynamic forces, cellular responses, and graft material properties at both anastomoses may uncover critical factors contributing to this asymmetric remodeling. Furthermore, modifying our PC-BU material to achieve a slower degradation profile may help determine whether a more gradual tissue transition improves outcomes. However, as evidenced by materials used in the medical field for decades, such as polylactic acid (PLA), polyglycolic acid (PGA), and polycaprolactone (PCL), which contain crystalline domains and degrade slowly, prolonged persistence of these materials can trigger adverse immunological responses[14,36,37]. Therefore, a deeper understanding of the mechanisms behind neointimal hyperplasia and venous stenosis is needed before modifying the base material or introducing targeted functionalizations to the graft design.

Advanced in silico models may further support this effort by simulating biomechanical and biochemical conditions, enabling the optimization of graft architecture to enhance long-term patency and clinical

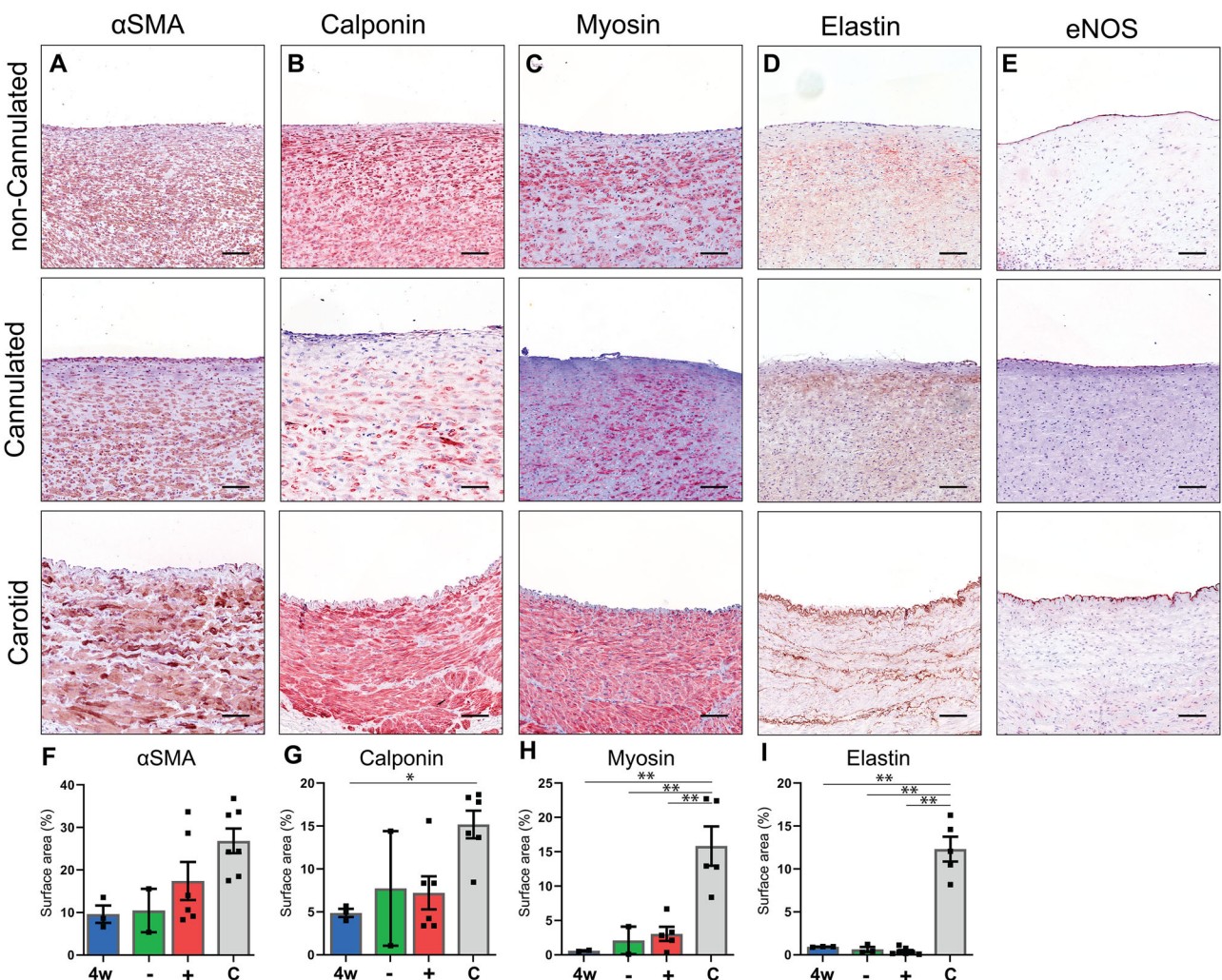

**Fig. 7 | Immunohistochemistry on vascular markers in representative transverse sections of the center part of the grafts, (non)-Cannulated explants at 12 weeks, and the native carotid artery as a control.** Stained with Novared® for smooth muscle marker αSMA (**A**), early contractile marker Calponin (**B**), mature contractile marker Myosin Heavy Chain (**C**), Elastin (**D**), and endothelial marker eNOS (**E**). With measured positive surface area of whole cross-sections of the grafts (**F–I**). 4 weeks (4w)(n = 3), Non-cannulated (−)(n = 2), Cannulated (+)(n = 6), Carotid Artery (**C**)(n = 7). Scale bars 50 μm. Data are shown as mean ± SEM. *$p < 0.05$, **$p < 0.01$. Kruskal–Wallis with Dunn's post-hoc (**F–I**).

outcomes[27,38]. These challenges must be addressed before progressing toward clinical translation. Future studies should include extended follow-up combined with high-frequency cannulation to evaluate structural and functional outcomes, assess potential risks such as aneurysm formation and infection, and develop clinically relevant protocols. Only after resolving these issues can cautious advancement toward patient application proceed, in accordance with applicable regulatory pathways.

In conclusion, this proof-of-concept study demonstrates the successful healing after repeated cannulation of a synthetic, biodegradable electrospun AV scaffold. This in vivo self-healing capacity was observed throughout the transformation of the scaffold into autologous, compliant vascular tissue, starting as early as 2 weeks after implantation. Consequently, TE vascular access grafts, due to their off-the-shelf availability and self-healing capacity, may provide long-term benefits over conventional, non-biodegradable, synthetic grafts.

## Methods
### Scaffold fabrication
The PC-BU and PCL-BU polymers (Fig. 1A) were synthesized following the procedures described previously[12–15,39]. 8 cm long grafts were created by integrating a 3D-printed coil between two layers of electrospun PC-BU scaffolds. Electrospun scaffolds were fabricated by dissolving PC-BU

polymer at a concentration of 23% w/v with a solvent mixture of 85% chloroform and 15% HFIP and were electrospun using an IME Technologies electrospinning machine. The intimal layer of PC-BU was electrospun with a solution flow rate of 70 μl/min⁻¹ while charged to 17 kV onto a 6 mm diameter stainless steel target in a climate-controlled chamber maintained at 23 °C and 55% humidity. The nozzle was surrounded with chloroform vapor using a controlled gas shield module with a chloroform flow rate of 60 μl/min to reduce solvent evaporation upon extrusion. The nozzle was translated from the chuck securing the target 90 mm along the length of the target at a speed of 10 mm/min, with a turn delay of 600 ms at both ends. The target was rotated at a speed of 250 rpm and charged from 0 kV to −3 kV over the course of 900 s from the activation of the high voltage. The intimal layer was spun for 16 min ± 30 s, with the scaffold thickness measured with a laser module while electrospinning, achieving a thickness of 450 ± 50 μm. Next, the 3D-printed spiral reinforcement was fabricated based on findings from a previous optimization on anti-kinking designs[40]. Using the Allevi 2 3D bioprinter, the spiral was printed directly onto the first electrospun layer, with an interspacing of 2.75 ± 0.25 mm at 60 °C. The adventitial layer of polymer was spun immediately after 3D printing with the same parameters as the intimal layer, but with a spinning time of 7 min ±30 s. Grafts were then placed in a vacuum overnight to evaporate any

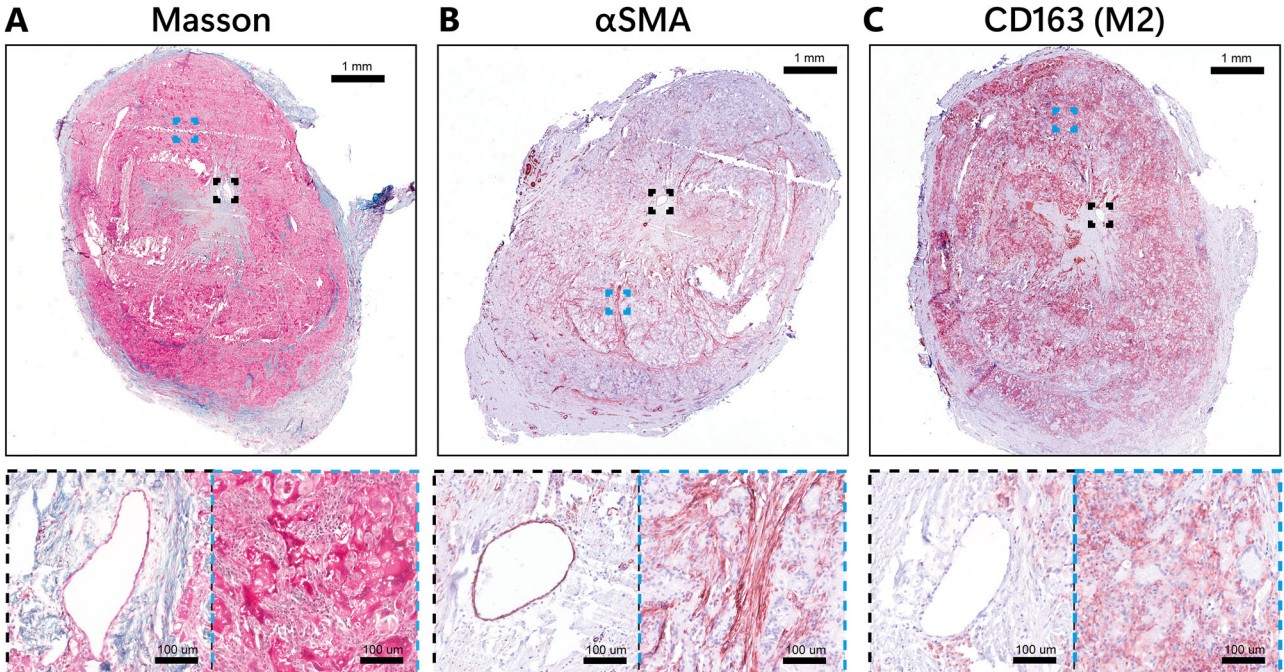

**Fig. 8 | Intimal Hyperplasia occurrence.** (Neo)Intimal Hyperplasia in representative transverse sections of the stenosed venous anastomotic region of the grafts, explants at 12 weeks. Masson's Trichrome (**A**) (cytoplasm (pink), Muscle fibers (red), collagen (blue), and nuclei (dark blue). Stained with Novared® for smooth muscle marker αSMA (**B**) and CD163, M2 macrophage marker (**C**). Zoomed inserts represent reduced lumen (Black) and neo-intima (Blue). Scale bars 1 mm and 100 μm for zoomed inserts.

remaining solvent, removed from the target, and stored in a 50 ml Falcon tube for transport. Prior to storage, a sample was cut from the end and washed twice in PBS and ultrapure water, followed by dehydration in a graded ethanol series. The samples were dried using a critical point dryer (CPD300, Leica, Austria) and visualized by SEM (Quanta 600F). Fiber diameter was measured from scanning electron microscopy images with ImageJ, with an average of 20 fibers per image.

**Differential scanning calorimetry**
Samples from the coil were cut and weighed (4–10 mg) directly into aluminum pans and hermetically sealed. Differential scanning calorimetry (DSC) was measured for these samples using a DSC Q2000 from TA Instruments, calibrated with an indium standard. Samples were heated to 180 °C at 40 °C/min, which marked the first heating run. Followed by two cooling and heating runs from −70 °C to 180 °C at 10 °C/min. The first and second heating runs were reported here, and the data were analyzed with Universal Analysis software (V4.5A, TA Instruments).

**Tensile testing of coil mixtures**
The samples used for the tensile tests consisted of pure polymer, 20 × 1 × 0.5 mm in size. The samples were tested with a Zwick/Roell Z010 tensile machine equipped with a 100 N load cell. First, the materials were stretched till a pre-load of 0.05 N was reached, followed by strain rate-controlled (10 mm/min) stretching while the force was measured (Fig. 1C). The elastic moduli of the materials were calculated by fitting a straight line through the graph in the elastic part.

**Animals**
Eight female Dutch milk goats (1.5–2 years old, ~65 kg) were housed under standard climate-controlled conditions in groups of up to six, with ad libitum access to food and water. Animals were acclimatized for seven days before the study. The study protocol was approved by the Animal Ethics Committee of Utrecht University (CCD - AVD1150020173344) and complied with Dutch animal experimentation laws.

To reduce animal use, bilateral implantations were performed. Implantation conditions (Sham, TE graft, ePTFE) were randomly assigned to both the animals and implantation sites (left/right). Additionally, a subset (n = 8) of bare scaffolds and all ePTFE controls underwent cannulation. Each animal received two different treatments (Supplementary Table 1).

An arteriovenous (AV) fistula was created between the internal jugular vein (IJV) and common carotid artery (CCA) using either a bare supramolecular polymer graft (n = 11) or an ePTFE graft (6 mm inner Ø, GORE® ACUSEAL, n = 3). Sham procedures (n = 2) served as controls (Fig. 9A). Animals were euthanized at either four weeks (n = 2) or twelve weeks (n = 6) post-implantation, and the grafts were harvested for further analysis.

**Surgical procedure**
Each animal received a buprenorphine patch (5 mg − 0.005 mg/h S.C.) 2 days before surgery. Prophylactic antibiotics consisting of amoxicillin with clavulanic acid were administered intravenously prior to the procedure (10 mg/kg I.V.). Animals were sedated with medetomidine (0.04 mg/kg I.M.), initiated and anesthetized with propofol (2 mg/kg I.V. bolus, followed by 10 mg/kg/h I.V. continuous infusion) and remifentanil (0.03 mg/kg/h I.V.). A midline incision in the neck was performed, and the IJV and CCA were carefully dissected bilaterally. Heparin (5000IU) was administered intravenously 3 min before clamping the IJV. Grafts were cut at a 45-degree angle at both ends and were anastomosed in an end-to-side fashion from the ipsilateral CCA distally to the IJV proximally using running 7-0 Prolene sutures (Ethicon #8683H) to create an AV conduit (Fig. 9B–D). The graft implantation was aligned with the vein's flow direction to minimize hemodynamic disturbances. Sham procedures were performed alongside graft implantation on the contralateral side, where vessels were dissected and clamped, without creating AV fistulas, serving to control for effects related to the surgical procedure itself. The duration of clamping was similar to the time used for the creation of the contralateral AV fistula. After removing the vascular clamps, vascular flow was confirmed in the jugular vein by a palpable thrill. In case of blood oozing through the graft wall, light gauze

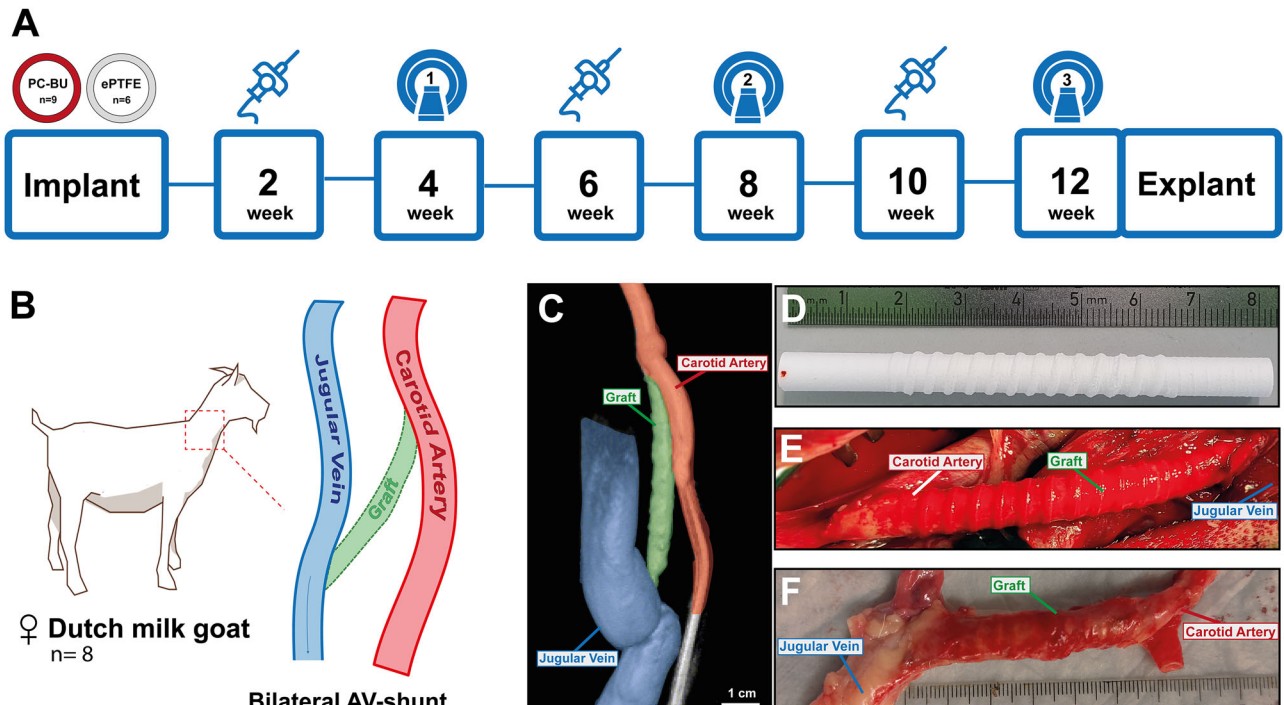

**Fig. 9 | Experimental set-up, implantation, and explantation.** Schematic overview of implantation, cannulation (weeks 2, 6, & 10), imaging (weeks 4, 8, & 12), and explantation after 12 weeks (**A**). 8 cm long grafts are implanted (**D**) between the carotid artery and jugular vein end-to-side in a straight configuration (**B**, **E**). Repeated 3D angiograms are taken to monitor patency (**C**). Implant (**D**) and explant after 12 weeks (**F**).

pressure or absorbable surgicel™ (Ethicon #1903) was used. The midline incision was closed using continuous 3.0 Vicryl subcutaneous sutures (Ethicon #V305H) and intracutaneous 3.0 Monocryl (Ethicon #W3650). For postoperative analgesia, a buprenorphine patch (5 mg–0.005 mg/h S.C) was given in combination with Meloxicam (0.5 mg/kg per day). Antibiotics, i.e., penicillin + dihydrostreptomycin (10 mg/kg I.M. per day), were given for 3 days. No anticoagulant or antiplatelet drugs were administered postoperatively.

## Cannulation

Grafts were cannulated at 2, 6, and 10 weeks after implantation (Fig. 9A). The subsequent time points, spaced approximately one month apart (2, 6 and 10 weeks), were selected to capture different phases of graft remodeling: 2 weeks representing the fully synthetic phase, 6 weeks representing the transitional phase during active remodeling, and 10 weeks representing a more matured, predominantly autologous tissue phase. Prior to cannulation, animals were sedated with medetomidine (0.04 mg/kg I.M.). Cannulation was performed by a skilled dialysis nurse guided by ultrasound with a 15 Gauge dialysis needle (Hospal #A15L15SGP+) attached to a line and a 50 mL Luer-Lock syringe (BD #309653). The cannulation site was targeted as close to the center of the graft as possible to ensure accurate cannulation and facilitate its localization in subsequent analyses. Aspiration of 50 mL of arterial blood was used to confirm needle entry into the graft lumen. Blood was then returned, the needle removed, and slight pressure was applied for a maximum of 5 min until hemostasis was achieved.

## Angiogram

A 3D angiogram was performed 4, 8, and 12 weeks after implantation using an Allura Xper FD20 device (Philips) (Fig. 9A). Each animal received a buprenorphine patch (5 mg − 0.005 mg/h S.C.) 2 days prior to imaging. Prophylactic antibiotics consisting of amoxicillin with clavulanic acid were administered intravenously prior to the procedure (10 mg/kg I.V.). Animals were sedated with medetomidine (0.04 mg/kg I.M.), initiated with and

anesthetized with propofol (2 mg/kg I.V. bolus, followed by 10 mg/kg/h I.V. continuous infusion) and remifentanil (0.03 mg/kg/h I.V.). A diagnostic catheter (7F HIGHFLOW(T)#527787 Cardinal Health) was inserted via the femoral artery and during imaging flowed with 60 ml (6 ml/s) contrast agent (Xenetix®, 350 mg I/ml). Images were analyzed and processed with Philips Dicom viewer R3.0 SP15. After the 12-week angiogram, animals were terminated with an overdose of pentobarbital natrium (500 mg/ml I.V.), and grafts were explanted and sectioned following Supplementary Fig. 1.

## Sample preparation and histology

For histology and immunohistochemistry, tissues were fixed in 4% formaldehyde, processed, and embedded in paraffin. Sections (3 μm thickness) were stained with Haematoxylin and Eosin (H&E), Masson's trichrome, and Picro Sirius red (SR) using standard techniques. Grafts were stained using Elastin (SAB4200718, 1:1000; Sigma), eNOS (610296, 1:200, BD Transduction Laboratories), VWF (ab6994, 1:400, Abcam), αSMA (ab7817,1:400, Abcam), Calponin (bs-0095R,1:400, BIOSS) and Myosin heavy chain (14-6400-82, Thermo). Sections were scanned and digitized with a NanoZoomer S360 (Hamamatsu C13220). Alternatively, for fluorescence, whole sections were imaged with a Leica DMi8 THUNDER Imager. Complete circumferential sections of the central portion of the graft were analyzed with CellProfiler[41]. Average wall and lumen diameter measurements were done on complete circumferential histology sections from the patent graft's arterial side, central region, and venous anastomosis.

## RNA Isolation, cDNA Synthesis, and qPCR

Samples for RNA analysis were snap-frozen in liquid nitrogen and stored at −80 °C. Total RNA was extracted using TRIzol® reagent (Invitrogen) and reverse transcribed from 200 ng of RNA using the Bioline SensiFAST cDNA Synthesis Kit (Bioline #BIO-65054). Quantitative analysis of goat genes (Supplementary Table 2) was performed using FastStart Universal SYBR Green (Roche #4913914001). B2M and 18S served as housekeeping genes, and relative mRNA expression levels were determined using the ΔCT method.

## Mechanical characterization of explanted grafts

From each explant, a 5 mm-long ring section was cut from the vessel for uniaxial tensile ring testing and SEM. raft sections were cut and gradually frozen in freezing medium (10% FCS, 10% DMSO in DMEM) in a Mr. Frosty™ and stored at −80 °C. Of each explant with an intact lumen, a segment 2–3 mm wide was prepared for testing. The dimensions of each segment were recorded with a calibrated digital microscope (VHC-500FE, Keyence). The remainder was fixed in 4% PFA for 30 min and stored in PBS at 4 °C for SEM imaging. The elastic moduli of the explants were determined by a uniaxial ring test (BioTester, 5 or 23 N load cell; CellScale) in combination with LabJoy software (V10.77, CellScale). For the ring tests, custom-made hooks of 1 mm in width were used. For each test, the zero-strain position was defined as 40% of the inner circumference of the graft segment. Each test consisted of 5 cycles of 10% incremental strain, up to 100% strain. All measurements were performed at a strain rate of 100% a minute. The modulus for each strain regime was calculated from the slope of the last 2.5% of the stress-strain curve of the last cycle.

## Scanning electron microscopy

Graft degradation was analyzed by scanning electron microscopy (SEM, Quanta 600F, FEI, Eindhoven). Neo-tissue was removed by 4.6% (v/v) sodium hypochlorite (Clorox) incubation for 15 min at room temperature. Afterwards, the tissues were washed twice in PBS and ultrapure water, followed by dehydration in a graded ethanol series. The samples were then dried using a critical point dryer (CPD300, Leica, Austria) and gold sputtered. They were visualized in high vacuum at 5 kV. Images were acquired at 500 × and 5000 × magnification.

## Statistics

Prior to commencing the animal experiments, a power calculation (α:0.05, β:0.90) was performed with G*power, a one-way t-test with two groups based on cell infiltration data in similar in vivo experiments in our group. Due to the small sample size and our primary concern being the preservation of statistical power rather than strict adherence to distributional assumptions, we did not conduct formal tests for normality. Data are presented as means ± standard error of the mean (SEM) and considered statistically significant at a $p < 0.05$. T-tests and one-way ANOVA with repeated measures (Kruskal–Wallis with Dunn's post-hoc) were used as appropriate, using GraphPad Prism 10 software (GraphPad Software).

## Data availability

The data that support the findings of this study are available from the corresponding author, M.C.V. upon reasonable request.

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

## Acknowledgements
The authors thank M. van de Kaa, B.C.T. van de Laar, K.R.D. Vaessen, DVM, N.J.M. Attevelt, and H.M.L. de Bruin for their assistance with the animal studies. K. den Ouden, P. M. de Bree for their assistance with the processing of the samples. This study was supported by a grant from ZonMw within the LSH 2Treat program, the Dutch Kidney Foundation [436001003 (InSiTeVx)], and the Gravitation Program "Materials Driven Regeneration", funded by the Netherlands Organization for Scientific Research (024.003.013).

## Author contributions
P.J.B., W.S., M.T., J.O.F., G.J.B., P.Y.W.D., C.V.C.B. and M.C.V.: Conception and design of the work; P.J.B., W.O., R.J.T., P.A.A.B., B.A., R.C.H.D., A.M.L. and H.C.B.: Data collection; P.J.B., W.O., P.A.A.B., B.A., R.C.H.D., A.M.L., H.C.B., J.O.F., M.C.V.: Data analysis and interpretation; P.J.B., J.O.F., M.T., C.V.C.B. and M.C.V.: Drafting the article; P.J.B., M.T., R.J.T., R.C.H.D., J.O.F., G.J.B., P.Y.W.D., C.V.C.B. and M.C.V.: Critical revision of the article.

## Competing interests
The authors declare no competing interests that may inappropriately influence or affect the integrity of the contents of the article. P.Y.W.D. is an inventor on patents that relate to the use of supramolecular interactions for advanced materials.

## Ethics approval
The study protocol was approved by the Animal Ethics Committee of the University of Utrecht (CCD - AVD1150020173344).
