## [Transparent Peer Review file · Communications Materials]

Evaluation of In Situ Tissue-Engineered Arteriovenous Grafts Suitable for Cannulation in a Large Animal Model

Corresponding Author: Dr Paul Besseling

Version 0:

Decision Letter:

Dear Dr Besseling,

Thank you for submitting your manuscript, "In situ Tissue Engineered Arteriovenous Grafts Display Self-Healing Capacity Upon Cannulation in a Large Animal Model", to Communications Materials. It has now been seen by 3 referees, whose comments are appended below. You will see that while they find your work of potential interest, they have raised substantial concerns that must be addressed. In light of these comments, we cannot accept the manuscript for publication, but are interested in considering a revised version that addresses these serious concerns.

In particular, Reviewer 3 suggests further experiments, while all Reviewers request further clarification on your experimental design.

We hope you will find the referees' comments useful as you decide how to proceed. Should further experimental data or analysis allow you to address these criticisms, we would be happy to look at a substantially revised manuscript. However, please bear in mind that we will be reluctant to approach the referees again in the absence of major revisions. If the revision process takes significantly longer than three months, we will be happy to reconsider your paper at a later date, as long as nothing similar has been accepted for publication at Communications Materials or published elsewhere in the meantime.

When submitting your revised manuscript, please include the following:

-A response letter with a point-by-point reply to each of the referee comments and a description of changes made. Please include the complete referee report in the response letter. Please note that the response letter must be separate to the cover letter to the editors.

-A marked-up version of the manuscript with all changes to the text in a different colored font. Please do not include tracked changes or comments. Please select the file type 'Revised Manuscript - Marked Up' when uploading the manuscript file to our online system.

-A clean version of the manuscript. Please select the file type 'Article File'.

-An updated [Editorial Policy](https://www.nature.com/documents/nr-editorial-policy-checklist.zip) checklist, uploaded as a 'Related Manuscript File' type. This checklist is to ensure your paper complies with all relevant editorial policies. If needed, please revise your manuscript in response to these points. Please note that this form is a dynamic 'smart pdf' and must therefore be downloaded and completed in Adobe Reader. Clicking this link will download a zip file containing the pdf.

Life Sciences: [Reporting requirements for life sciences research](https://www.nature.com/documents/nr-reporting-summary.zip)

Life Sciences Policy: Policy requirements for life sciences research

Please use the following link to submit your revised manuscript files:

Link Redacted

Please do not hesitate to contact me if you have any questions or would like to discuss the required revisions further. Thank you for the opportunity to review your work.

Best regards,

Steven Caliri, PhD
Editorial Board Member
Communications Materials
orcid.org/0000-0002-7506-3079

Reviewers' comments:

Reviewer #1 (Remarks to the Author):

In this manuscript, the authors used PC-BU to construct a three-layered vascular graft (TE graft), enhancing the outer layer of the vascular graft with an electrospun layer and using helical fibers as a sandwich core to prevent structural damage as previous report. The TE graft exhibits vascular regeneration capabilities and resistance to repeated punctures. The work holds significant promise for clinical applications. However, using helical structures to enhance anti-kinking properties has been widely applied, and the three-layered blood vessels have also been previously reported. Thus, the innovation of materials design is not particularly prominent. Additionally, there are some perplexing aspects in the authors' research, detailed as follows:

1. The authors established a sham (n=2) group as a control. In which data were samples from the sham group applied? The author showed some data of carotid, were veins treated with sham used for comparative analysis?
2. On page 11, line 248, the authors stated the thickness and spacing of the 3D-printed anti-kinking helical rings, but this data could not be confirmed in the scanning electron microscopy image of Fig. 1B.
3. On page 11, line 252, the authors claimed no hold were observed. I'm wondering under what bending conditions (for example, the bending angle and whether the graft subjected to internal pressure) were the tests conducted? We suggest the authors to conduct a quantitative assessment of the anti-kinking performance. Besides the helical design improving anti-kinking characteristics, the elasticity of the material itself also plays a role. The authors should compare the anti-kinking properties of different 80:20 w/w PCL-BU:PCL mixtures, pure PCL-BU, PC-BU, and PCL. We also suggest the authors provide stress-strain curves for blood vessels composed of different materials.
4. What is the reason the authors did not consider using the 80:20 w/w PCL-BU:PCL mixture and pure PCL-BU for in vivo animal testing?
5. On page 12, line 286, occlusion was predominantly caused by stenosis at the venous anastomosis (n=6/8). In which groups did the six occlusions occur? Did they affect the patency of the blood vessels?
6. From Fig. 3A, it appears that the luminal surface of the TE graft has significantly decreased. The authors should provide quantitative analysis with ultrasound or histological staining results.
7. In Fig. 3C, the wall thickness of the blood vessel at 12 weeks appears to have significantly increased. What could be the reason? Is the increase in the wall thickness due to intimal hyperplasia?
8. In Fig. 6, the lumen of the TE graft at 12 weeks appears to be more dilated, which seems inconsistent with the macroscopic photo (Fig. 3A). The authors need to check the relevant data or scales to clarify the changes in the diameter of the blood vessels in all animals.
9. The authors conducted histological and mechanical evaluations on blood vessels with or without cannulation. It seems that puncturing every two weeks cannot achieve coverage of the entire area of the graft. How did the authors ensure in the pathological analysis that the sections or samples used for mechanical assessment were from the punctured areas?
10. The authors analyzed that the reason for the decreased patency rate of the TE Graft may be related to the stenosis at the venous anastomosis. What is the probability of this occurring in ePTFE and TE grafts, separately?
11. How is the degradation of the helical core in the TE Graft after 12 weeks post-implantation? What are the mechanical changes of the TE Graft after 12 weeks compared to before implantation? Is there a risk of aneurysm formation during longer-term vascular remodeling?

Reviewer #2 (Remarks to the Author):

The manuscript "In situ Tissue Engineered Arteriovenous Grafts Display Self-Healing Capacity Upon Cannulation in a Large Animal Model" provides valuable insights into the development of biocompatible grafts and was very interesting to read. Below are comments suggesting areas where further clarification or additional information might be helpful.

Purpose of Grouping:

Since no intergroup comparisons were performed among the four experimental groups, the rationale for creating these groups seems somewhat unclear. Could you please explain the purpose of this grouping and why no intergroup comparisons were conducted?

Details of Animal and Graft Allocation:

The allocation of animals and grafts among Groups 1 through 4 is not clearly explained. Indicating how many animals were assigned to each group would help clarify the experimental setup. Specifically, while the manuscript mentions 11 TE grafts, 3 ePTFE grafts, and 2 sham models, the composition of each group is not explicitly described. Including a table or another visual representation might improve clarity. Additionally, were all three ePTFE grafts cannulated after implantation? If any ePTFE grafts were left uncannulated, could you specify how many?

Postoperative Antithrombotic Therapy:

The manuscript states that heparin was administered intraoperatively, but there is no mention of whether antiplatelet or anticoagulant drugs were administered postoperatively. Since graft occlusion can result from both intimal hyperplasia and early postoperative thrombosis, could you clarify whether postoperative antithrombotic therapy was provided? If administered, details such as drug names, dosages, and treatment durations would be appreciated.

Graft Length Information:

While the inner diameters of the TE grafts (5.9 ± 0.2 mm) and ePTFE grafts (6 mm) are provided, the actual lengths of the implanted grafts are not mentioned. Including the implanted graft lengths for both types would enhance clarity.

Patency Rate Calculations:

The manuscript states patency rates of 70% at 4 weeks and 50% at 12 weeks, but the calculation criteria (numerator and denominator) are unclear. At 4 weeks, with 11 TE grafts involved, how was the 70% patency rate calculated? Clarifying the numbers of occluded and patent grafts would be helpful. Additionally, could you specify which groups the two animals euthanized at 4 weeks belonged to?

Patency Evaluation Timing:

Cannulation of TE and ePTFE grafts occurred at 2, 6, and 10 weeks post-implantation. Successful reverse blood flow upon cannulation would indicate patent grafts. While patency was evaluated by CT at 4, 8, and 12 weeks, would data on graft patency from cannulations at 2, 6, and 10 weeks be available? This information could provide additional insights. For example, if a graft was patent at 8 weeks but occluded at 12 weeks, was cannulation at 10 weeks successful? Similarly, if a graft was occluded at 4 weeks, did thrombosis already occur by the first cannulation at 2 weeks, preventing reverse blood flow?

Regarding the Cause of Occlusion:

The discussion mentions intimal hyperplasia as the cause of graft occlusion; however, specific evidence such as CT images, macroscopic findings of the explanted grafts, or histological data from the occluded grafts does not appear to be presented. Providing visual data on the occluded grafts could strengthen the discussion and enhance the manuscript's conclusions.

Stenosis Evaluation:

Some patent grafts might still have experienced stenosis due to intimal hyperplasia. Were all patent grafts entirely free of intimal hyperplasia? If any degree of stenosis was observed, could the severity be described with supporting histological or imaging evidence?

Comparison Based on Cannulation:

The manuscript does not clearly describe how patency rates or wall thickness were evaluated based on whether grafts were cannulated. Did cannulation influence intimal hyperplasia development or patency rates? Further data and discussions would be appreciated.

Wall Thickness Measurement Locations:

The manuscript does not specify where wall thickness measurements were taken (e.g., arterial anastomosis, graft center, venous anastomosis). Supplementary data suggest that the graft's arterial side, central region, and venous anastomosis were sampled. Could you clarify whether the reported wall thickness represents an average of these measurements? Were both patent and occluded grafts considered in this analysis?

Sample Sources for Mechanical Testing:

The origins of samples used for mechanical testing in Figure 4 ($n=1$ at 4 weeks, $n=7$ at 12 weeks, and $n=2$ for ePTFE) are unclear. Were multiple samples cut from the same grafts for these tests? Additionally, were only patent grafts tested, or were occluded grafts also included? Clarifying these sample sources would be beneficial.

Figure Suggestion - Arrow Annotation in Figure 6:

The manuscript states, "Irrespective of the graft type, enlarged outflow jugular veins with stenosis around the venous anastomoses were observed, which did not resolve over time (Figure 6)." Adding arrows to indicate this region in the CT images of Figure 6 might enhance the figure's clarity.

Consideration of Graft Occlusion Factors:

In this study, the TE graft coil was improved, enhancing graft compliance and preventing coil fragmentation after implantation. However, despite these improvements, there was no clear enhancement in graft patency. The authors discussed intimal hyperplasia as the primary cause of graft occlusion, but there may be room for further consideration of other potential contributing factors. For example, thrombosis, perivascular tissue reactions, or mechanical stress at the anastomosis sites could be relevant. Addressing these aspects in the discussion might provide a more comprehensive interpretation of the study's findings.

Reviewer #3 (Remarks to the Author):

Reviewer Comments

1. Self-healing Mechanism

How is self-healing ability quantitatively assessed, such as the time course of healing post-puncture and collagen deposition? The manuscript lacks detailed quantitative data. Please include these key metrics.

What are the cellular mechanisms involved in tissue regeneration? The manuscript does not clearly discuss cellular processes like cell proliferation, migration, and differentiation, which are critical to explaining the self-healing mechanism. Additional data should be provided to support these mechanisms.

Are there quantitative data on tissue reconstruction, particularly regarding collagen deposition and new tissue formation at different time points? The manuscript discusses tissue reconstruction but lacks these quantitative data. Please add them to better illustrate the healing process.

2. Material Design and Characterization

What is the rationale for the 60:40 PCL-BU:PCL ratio? Has the effect of different ratios on self-healing ability been considered? Please clarify the choice of this ratio and explore whether other ratios were tested and considered.

How were the electrospinning parameters optimized? The manuscript does not provide details on how parameters such as voltage, flow rate, and solution concentration were optimized, and how these affect the material's structure and functionality. Please include this information.

How was the spiral pitch in the 3D-printed scaffold determined? The manuscript does not discuss the rationale behind the selection of spiral pitch and its impact on the graft's mechanical properties and vascular regeneration. Please explain the choice and its effect on graft functionality.

3. Experimental Design

What are the criteria for puncture site selection? The manuscript does not clarify the criteria for selecting puncture sites or how different locations might impact self-healing ability. Please explain the selection process and discuss any location-dependent effects on graft performance.

Why were the time points (2, 6, 10 weeks) selected for analysis? The manuscript lacks explanation of why these specific time points were chosen. Please clarify the relationship between these time points and self-healing ability.

4. Clinical Translation

Is the puncture frequency comparable to clinical practice in hemodialysis patients? The manuscript does not compare puncture frequency with real clinical conditions, particularly the long-term impact of frequent punctures on graft performance. Please discuss the graft's suitability for clinical use, especially regarding repeated punctures.

Have strategies for preventing venous stenosis, such as drug coatings or other treatments, been considered to optimize long-term patency? The manuscript does not mention such strategies. Please discuss these strategies and their potential role in maintaining graft patency over time.

How will infection risk be controlled in clinical applications? The manuscript does not address infection risk, especially given the need for frequent punctures in hemodialysis access. Please consider the use of surface coatings, antimicrobial materials, or other preventive measures and discuss them.

5. Data Presentation and Analysis

5.1. Technical Details:

The scale bars in SEM images in Figures 4D and 4F are inconsistent. Please unify the scale bars.

The statistical methods for wall thickness data in Figure 3C are not specified, and the post-hoc testing method for mechanical performance in Figure 4I is missing. The methods for normality testing in Figures 8F-I are also incomplete. Please provide the missing statistical details.

The meaning of the yellow arrows in Figure 6 is unclear, and the specific markers used for tissue staining in Figure 7 need more detailed explanation.

5.2. Data Presentation:

CT imaging data lack quantitative analysis. Please provide quantitative results such as changes in puncture site morphology or lumen diameter.

The mechanical testing data are mainly presented for certain time points. It would be beneficial to include data from additional time points to assess the trend in mechanical performance over time.

The presentation of tissue reconstruction lacks clear time-series data. Please present data across more time points to illustrate cellular proliferation, collagen deposition, and other processes involved in tissue healing.

6. Suggested Additional Experiments

6.1. Material Characterization:

Before selecting the 60:40 PCL-BU:PCL ratio for implantation, please perform a comparison of different ratios using SEM to strengthen the justification for the chosen ratio.

Perform contact angle experiments with both water and blood to compare their interaction with the graft surface.

6.2. Biological Analysis:

Supplementary immunofluorescence staining for tissue markers, particularly α SMA and endothelial markers, is needed to assess vascular regeneration.

In vitro biocompatibility studies, including cell proliferation and apoptosis assays, as well as immunofluorescence staining (e.g., CD31, α SMA), should be conducted.

In vitro blood compatibility experiments should assess platelet adhesion and thrombus formation potential.

6.3. Data Analysis:

Strengthen statistical analysis of existing data and conduct quantitative image analysis where applicable.

Suggestions for Revision

1. Expand Discussion:

Theoretical basis for material selection: The manuscript lacks a thorough explanation of why PC-BU and PCL were chosen. Please provide a detailed discussion of the material selection rationale.

Comparison with similar materials in the literature: The manuscript lacks a detailed comparison with other common materials (e.g., ePTFE, modified PCL). Please include this comparison to highlight the advantages of the chosen materials.

Clinical feasibility: The manuscript lacks a comprehensive analysis of clinical translation, particularly addressing potential issues such as puncture frequency, infection control, and graft durability in clinical settings.

Limitations of animal models: The limitations of the animal models used, particularly in simulating clinical pathophysiological conditions, should be discussed in more detail.

Future improvement strategies: The manuscript does not propose specific strategies for future improvements in materials, processing, or clinical application. Please include a discussion of future research directions.

2. Improve Data Presentation:

Include detailed statistical methods for all analyses.

Improve figure quality and annotations to ensure clarity and consistency in presenting data.

3. Strengthen Clinical Relevance:

Discuss the challenges faced in clinical translation.

Compare the proposed solution with current clinical standards.

Outline potential regulatory pathways for clinical implementation.

Conclusion

This study offers a promising solution for vascular access in hemodialysis patients. However, several aspects of the manuscript require further clarification and improvement. The authors are encouraged to revise the manuscript according to the suggestions provided, adding relevant experiments, data analysis, and discussions. This will significantly enhance the reliability of the conclusions and strengthen the contribution to the field of vascular tissue engineering.

<http://www.nature.com/authors> for information about policies, services and author benefits**

Communications Materials is committed to improving transparency in authorship. As part of our efforts in this direction, we are now requesting that all authors identified as 'corresponding author' create and link their Open Researcher and Contributor Identifier (ORCID) with their account on the Manuscript Tracking System prior to acceptance. ORCID helps the scientific community achieve unambiguous attribution of all scholarly contributions. You can create and link your ORCID from the home page of the Manuscript Tracking System by clicking on 'Modify my Springer Nature account' and following the instructions in the link below. Please also inform all co-authors that they can add their ORCIDs to their accounts and that they must do so prior to acceptance.

Version 1:

Decision Letter:

Dear Dr Besseling,

Thank you for submitting your manuscript, "Evaluation of In Situ Tissue-Engineered Arteriovenous Grafts Suitable for Cannulation in a Large Animal Model", to Communications Materials. It has now been seen again by 3 referees, whose comments are appended below. You will see that while Reviewers 1 and 2 find your work acceptable, some important points are still raised by Reviewer 3. We remain interested in the possibility of publishing your study in Communications Materials, but would like to consider your response to these concerns in the form of a revised manuscript before we make a decision on publication.

In particular, Reviewer 3 requests further clarification and discussion.

We therefore invite you to revise and resubmit your manuscript, taking into account the points raised.

When submitting your revised manuscript, please include the following:

-A response letter with a point-by-point reply to each of the referee comments and a description of changes made. Please include the complete referee report in the response letter. Please note that the response letter must be separate to the cover letter to the editors.

-A marked-up version of the manuscript with all changes to the text in a different colored font. Please do not include tracked changes or comments. Please select the file type 'Revised Manuscript - Marked Up' when uploading the manuscript file to our online system.

-A clean version of the manuscript. Please select the file type 'Article File'.

Please ensure that the following requirements are met, and that any other relevant checklists are completed and uploaded under the 'Related Manuscript file' type with the revised article.

Chemical and biomolecular materials: [Characterization of chemical and biomolecular materials](https://www.nature.com/commschem/submit/submission-guidelines#characterization)

Life sciences reporting summary: [Reporting requirements for life sciences research](https://www.nature.com/documents/nr-reporting-summary.pdf)

In the event that your manuscript is accepted we will provide detailed guidance on our journal policies and formatting. You may however wish to ensure that the manuscript complies with our house style at this stage. See our style and formatting guide (<https://www.nature.com/documents/commsj-phys-style-formatting-guide-accept.pdf>) and checklist (<https://www.nature.com/documents/commsj-phys-style-formatting-checklist-article.pdf>) for reference.

Data availability statements and data citations policy: All Communications Materials manuscripts must include a section titled "Data Availability" at the end of the Methods section or main text (if no Methods). More information on this policy, and a

list of examples, is available at <http://www.nature.com/authors/policies/data/data-availability-statements-data-citations.pdf>.

- Accession codes for deposited data
- Other unique identifiers (such as DOIs and hyperlinks for any other datasets)
- At a minimum, a statement confirming that all relevant data are available from the authors
- If applicable, a statement regarding data available with restrictions
- If a dataset has a Digital Object Identifier (DOI) as its unique identifier, we strongly encourage including this in the Reference list and citing the dataset in the Data Availability Statement.

DATA SOURCES: We strongly encourage authors to deposit all new data associated with the paper in a persistent repository where they can be freely and enduringly accessed. We recommend submitting the data to discipline-specific, community-recognized repositories, where possible and a list of recommended repositories is provided at <http://www.nature.com/sdata/policies/repositories>.

If a community resource is unavailable, data can be submitted to generalist repositories such as [figshare](https://figshare.com/) or [Dryad Digital Repository](http://datadryad.org/). Please provide a unique identifier for the data (for example a DOI or a permanent URL) in the data availability statement, if possible. If the repository does not provide identifiers, we encourage authors to supply the search terms that will return the data. For data that have been obtained from publicly available sources, please provide a URL and the specific data product name in the data availability statement. Data with a DOI should be further cited in the methods reference section.

Please use the following link to submit your documents:

Link Redacted

We hope to receive your revised paper within six weeks; please let us know if you aren't able to submit it within this time so that we can discuss how best to proceed. If we don't hear from you, and the revision process takes significantly longer, we will close your file. In this event, we will still be happy to reconsider your paper at a later date, as long as nothing similar has been accepted for publication at Communications Materials or published elsewhere in the meantime.

Please do not hesitate to contact me if you have any questions or would like to discuss these revisions further. We look forward to seeing the revised manuscript and thank you for the opportunity to review your work.

Best regards,

Steven Caliarì, PhD
Editorial Board Member
Communications Materials
orcid.org/0000-0002-7506-3079

Reviewers' comments:

Reviewer #1 (Remarks to the Author):

The authors have addressed all my comments.

Reviewer #2 (Remarks to the Author):

The authors have responded carefully to all of my previous comments, and the revised manuscript shows clear improvement. The experimental framework is now more clearly described, the methodology is more transparent, and the discussion has been strengthened.

The manuscript reads clearly, and I did not find any significant grammatical or typographical errors.

I consider the manuscript suitable for publication in its current form and have no further concerns.

Reviewer #3 (Remarks to the Author):

Reviewer's Second Comments

The authors have provided a thorough and thoughtful response to the initial review, significantly improving the manuscript's clarity and depth. They have added gene expression data (Col1, Col3, α SMA), detailed the 60:40 PCL-BU:PCL rationale with stress-strain curves, fully described electrospinning/3D-printing parameters, and expanded discussion on clinical translation and model limitations. However, several points still require clarification or further detail:

1. Quantitative Assessment of Self-Healing

The authors explain that puncture sites could not be clearly identified at explant and have refocused on neo-tissue formation and cannulation capability. This is reasonable, but please clarify in the Discussion why puncture sites could not be tracked post-explanations (e.g., material properties, healing kinetics, ultrasound guidance). A clear statement of these limitations will help readers understand why quantitative healing data are absent.

2. In Vivo Impact of the 60:40 PCL-BU:PCL Ratio

The mechanical data for the 60:40 blend are convincing in vitro. Please discuss how this ratio specifically influenced vessel remodeling and mechanical stability in the goat model. Cite any observations (e.g., reduced kinking, consistent patency) that demonstrate real-world benefits.

3. Proof-of-Concept Emphasis

You correctly acknowledge that cannulation frequency differs from clinical practice. Please further emphasize in both the Introduction and Discussion that this study is a proof-of-concept, and that high-frequency cannulation, long-term fatigue, stenosis, and infection will be key areas for future work.

4. Data Presentation and Analysis

Technical Details: All scale-bar inconsistencies have been fixed. The choice of non-parametric statistics is now properly justified given the sample size. Figure annotations (arrows in Figs. 6–7) are now clear.

Quantitative Data: CT puncture sites remain undetectable, but lumen diameter quantification is now included—this is satisfactory. The inability to increase mechanical testing time points is fully explained. The addition of gene expression (Supplemental Fig. 3) substantially strengthens the remodeling analysis.

Recommendation

The authors have addressed most reviewer concerns in detail and the manuscript has improved substantially. To further strengthen its rigor and transparency, please:

1. Clarify the reasons for the absence of puncture-site tracking and quantitative healing metrics.
2. Emphasize the proof-of-concept nature of this study and clearly outline high-frequency cannulation and long-term performance as future work.

After these refinements, the manuscript will be suitable for publication in Communications Materials.

Communications Materials is committed to improving transparency in authorship. As part of our efforts in this direction, we are now requesting that all authors identified as 'corresponding author' create and link their Open Researcher and Contributor Identifier (ORCID) with their account on the Manuscript Tracking System prior to acceptance. ORCID helps the scientific community achieve unambiguous attribution of all scholarly contributions. You can create and link your ORCID from the home page of the Manuscript Tracking System by clicking on 'Modify my Springer Nature account' and following the instructions in the link below. Please also inform all co-authors that they can add their ORCIDs to their accounts and that they must do so prior to acceptance.

Version 2:

Decision Letter:

Dear Dr Besseling,

Thank you for resubmitting your manuscript titled "Evaluation of In Situ Tissue-Engineered Arteriovenous Grafts Suitable for Cannulation in a Large Animal Model". I am delighted to say that we are happy, in principle, to publish a suitably revised version in Communications Materials.

We therefore invite you to edit your manuscript to comply with our journal policies and formatting style in order to maximise the accessibility and therefore the impact of your work.

EDITORIAL REQUESTS

* Your manuscript should comply with our policies and format requirements, detailed in our style and formatting guide (<https://www.nature.com/documents/commsj-phys-style-formatting-guide-accept.pdf>).

* Please edit your manuscript according to the editorial requests in the attached table, and outline revisions made in the right hand column. If you have any questions or concerns about any of our requests, please do not hesitate to contact me. It is important that each request be addressed in order to avoid delays in accepting your manuscript. Please upload the completed table with your manuscript files as a Related Manuscript file.

* The editorial requests table also includes a full list of the files that must be provided upon resubmission. Please upload your files according to this table.

Nature journals require authors of life sciences research papers to include relevant details about several elements of experimental and analytical design in their manuscripts. This initiative aims to improve the transparency of reporting and the reproducibility of published results and is described at: <http://www.nature.com/authors/policies/reporting.pdf>. To ensure that your manuscript complies with our policy, please pay close attention to the 'methods' and 'legends' sections of our checklist for authors: <https://www.nature.com/documents/nr-reporting-summary.pdf> Reporting requirements for life sciences research. You may also find the following collection of articles on statistics for biologists helpful: <http://www.nature.com/collections/qghhqm> Statistics for Biologists.

OPEN ACCESS

Communications Materials is a fully open access journal. Articles are made freely accessible on publication. For further information about article processing charges, open access funding, and advice and support from Nature Research, please visit <https://www.nature.com/commsmat/open-access>

Please use the following link to submit your revised files:

Link Redacted

We hope to hear from you within two weeks; please let us know if the process may take longer.

Best regards,

Dr Jet-Sing Lee
Senior Editor
Communications Materials
